# Sosuga Virus Detected in Egyptian Rousette Bats (*Rousettus aegyptiacus*) in Sierra Leone

**DOI:** 10.3390/v16040648

**Published:** 2024-04-22

**Authors:** Brian R. Amman, Alusine H. Koroma, Amy J. Schuh, Immah Conteh, Tara K. Sealy, Ibrahim Foday, Jonathan Johnny, Ibrahim A. Bakarr, Shannon L. M. Whitmer, Emily A. Wright, Aiah A. Gbakima, James Graziano, Camilla Bangura, Emmanuel Kamanda, Augustus Osborne, Emmanuel Saidu, Jonathan A. Musa, Doris F. Bangura, Sammuel M. T. Williams, George M. Fefegula, Christian Sumaila, Juliet Jabaty, Fatmata H. James, Amara Jambai, Kate Garnett, Thomas F. Kamara, Jonathan S. Towner, Aiah Lebbie

**Affiliations:** 1Centers for Disease Control and Prevention, Viral Special Pathogens Branch, 1600 Clifton Rd. NE, Atlanta, GA 30329, USA; wuc2@cdc.gov (A.J.S.); tss3@cdc.gov (T.K.S.); evk3@cdc.gov (S.L.M.W.); ikw6@cdc.gov (J.G.); 2Department of Biological Sciences, Njala University, Njala, Sierra Leone; alusinehkoroma@gmail.com (A.H.K.); immahsoso1909@gmail.com (I.C.); ifoday@njala.edu.sl (I.F.); jojnya@gmail.com (J.J.); bakarribrahim@gmail.com (I.A.B.); camillabangura2@gmail.com (C.B.); emmanuelskamanda@gmail.com (E.K.); augustusosborne2@gmail.com (A.O.); emanuelsaidu@gmail.com (E.S.); jonathanamusa@gmail.com (J.A.M.); dorisbangs8@gmail.com (D.F.B.); samuelmaxwellwilliams@gmail.com (S.M.T.W.); georgefefegula@gmail.com (G.M.F.); sumailachristian@gmail.com (C.S.); julietjabaty2012@yahoo.com (J.J.); hajafatmatajames@gmail.com (F.H.J.); 3Oak Ridge Institute Science and Education, Oak Ridge, TN 37830, USA; esm0@cdc.gov; 4National Public Health Agency, Wilberforce, 42A Main Mottor Road, Freetown, Sierra Leone; gbakimaaa2009@gmail.com; 5Ministry of Health and Sanitation, Brookfields, Youyi Building, Freetown, Sierra Leone; amarajambai@yahoo.com; 6Ministry of Agriculture and Forestry, Brookfields, Youyi Building, Freetown, Sierra Leone; majelarnett@yahoo.co.uk; 7National Protected Area Authority, 4-6 FA John Avenue, Main Congo Town Road, Freetown, Sierra Leone; tfkamara01@gmail.com

**Keywords:** Sosuga virus, paramyxovirus, *Rousettus aegyptiacus*, Egyptian rousette, range extension, zoonotic viruses, viral zoonoses, disease ecology

## Abstract

Sosuga virus (SOSV), a rare human pathogenic paramyxovirus, was first discovered in 2012 when a person became ill after working in South Sudan and Uganda. During an ecological investigation, several species of bats were sampled and tested for SOSV RNA and only one species, the Egyptian rousette bat (ERBs; *Rousettus aegyptiacus*), tested positive. Since that time, multiple other species have been sampled and ERBs in Uganda have continued to be the only species of bat positive for SOSV infection. Subsequent studies of ERBs with SOSV demonstrated that ERBs are a competent host for SOSV and shed this infectious virus while exhibiting only minor infection-associated pathology. Following the 2014 Ebola outbreak in West Africa, surveillance efforts focused on discovering reservoirs for zoonotic pathogens resulted in the capture and testing of many bat species. Here, SOSV RNA was detected by qRT-PCR only in ERBs captured in the Moyamba District of Sierra Leone in the central region of the country. These findings represent a substantial range extension from East Africa to West Africa for SOSV, suggesting that this paramyxovirus may occur in ERB populations throughout its sub-Saharan African range.

## 1. Introduction

Sosuga virus (SOSV), is a member of the large and diverse virus family *Paramyxoviridae* (subfamily *Rubulavirinae*, genus *Pararubulavirus*). This virus was first identified in clinical samples obtained from an infected wildlife biologist who had recently been working in the field capturing and sampling bats in South Sudan and Uganda [1]. There was no onward transmission of the virus. Subsequent testing of tissues from several species of bats collected from multiple locations in Uganda, identified Egyptian rousette bats (ERBs: Family Pteropodidae, *Rousettus aegyptiacus*) as the only species of bat to test positive for SOSV RNA by qRT-PCR [2]. 

Numerous bat species have been identified as hosts of paramyxoviruses [3] since the isolation of a bat-associated parainfluenza virus from a Leschenault’s rousette bat (*Rousettus leschenaultii*) in 1966 [4] and Mapuera virus from a little yellow-shouldered bat (*Sturnira lilium*) in 1979 [5]. Like SOSV, many paramyxoviruses are known to be promiscuous and are pathogenic to a variety of species of animals, including humans [1,3,6,7,8]. Bats from the chiropteran family Pteropodidae are also known to host the human pathogenic paramyxoviruses Nipah virus (NiV) and Hendra virus (HeV) [9,10,11,12]. Moreover, recent studies report that captive-bred ERBs could be experimentally infected with SOSV with no overt signs of morbidity, mild pathology, and shed the virus in urine, feces, and saliva, suggesting they are at least competent hosts and a potential reservoir for this human pathogenic paramyxovirus [13,14].

ERBs have a fragmented distribution across Sub-Saharan and South Africa and can form large, dense colonies numbering up to and over 100,000 bats [15,16,17] (Figure 1). They typically breed twice a year in the tropical biomes, producing up to thousands of juvenile bats every six months [18,19]. Field studies in Uganda showed 4.7% (62/1331) of all ERBs tested to be actively infected with SOSV [2]. At one location (Kitaka Mine), the active infection rate for SOSV in ERBs was as high as 10.2% (41/400), possibly due to the colony undergoing repopulation after an extermination attempt to eliminate Marburg virus (MARV)-infected ERBs from the mine [20]. Until now, these have been the only reported occurrences of SOSV in wild caught bats, although other rubulaviruses have been identified in ERBs in South Africa [21]. Recently, tissue, blood and swab samples from mutliple species of bats have been tested as part of a larger filovirus surveillance effort in Sierra Leone [22] following the 2014 Ebola virus outbreak. Of all the species tested from this region, only the ERB samples had detectable SOSV RNA. This is the first identified occurrence of SOSV in bats outside of East Africa (Uganda) and the first reported occurrence of this human pathogenic paramyxovirus in West Africa (Sierra Leone). 

## 2. Materials and Methods

### 2.1. Bat Capture and Processing

All the work described in this study was performed as a collaboration between Njala University, Sierra Leone, and the United States Centers for Disease Control and Prevention (CDC). All bat captures, sampling and testing procedures were performed with the permissions from The Ministry of Agriculture, Forestry, and Food Security and with approval of the Centers for Disease Control and Prevention Institutional Animal Care and Use Committees (IACUC; protocol number: 2943AMMMULX). The chiropteran taxonomy used in this manuscript follows that of Wilson and Reeder [23]. Bat captures were performed following methods previously described [22]. Briefly, ERBs were captured using mist nets placed at cave openings (Kasewe Cave, Moyamba District) or in suitable habitat and natural flyways and corridors (Tailu Village, Kailahun District; Kangari Hills Forest Reserve, Bo District; Figure 1). The majority of bat captures occurred during the fall (September–November, 2017–2020) with one capture event occurring in January 2016 and one in February 2021. Captured bats were placed in breathable cotton bags and transported to a processing site where they were processed via complete necropsy following procedures outlined previously [24]. Captured bats were humanely euthanized under anesthesia whereupon a cardiac blood sample was obtained. Polyester-tipped swabs were used to collect oral secretion samples (*n* = 2) and then placed in a virucidal lysis buffer (MagMax–Life Technologies) for PCR analysis and in viral transport media (VTM: 0.5 mL aliquots of Dulbecco’s Modified Eagle Medium containing 2% heat-inactivated fetal bovine serum, 100 units/mL penicillin, 100 μg/mL streptomycin, 50 μg/mL gentamicin, and 2.50 μg/mL amphotericin B) for virus isolation. Visceral tissues (liver, spleen, axillary lymph node, salivary gland, and colon) were collected from all captured bat species during necropsies and either flash frozen in liquid nitrogen for storage or placed in virucidal lysis buffer for inactivation and downstream PCR analysis. No bat species captured and sampled in this study were classified as either threatened or endangered.

### 2.2. Statistical Analyses

All statistical analyses were performed using SPSS Statistics v29.0.1.0 (IBM Corp, Armonk, NY, USA). The ecological data collected at the trapping sites were analyzed for age, sex, and capture bias using two-sided Pearson’s chi-squared tests of independence. Adjusted standardized residuals (z-scores) were calculated and then compared against the critical z-value (61.96) for α = 0.05. 

### 2.3. Sosuga Virus qRT-PCR

Nucleic acid was extracted on the MagMAX Express-96 Deep Well Magnetic Particle Processor (Thermo Fisher Scientific, Waltham, MA, USA) from tissue homogenates using the MagMAX Total RNA Isolation Kit (Thermo Fisher Scientific) and oral swabs using the MagMAX Pathogen RNA/DNA Kit (Thermo Fisher Scientific) at Njala University, as reported previously [22]. SOSV N (nucleocapsid) gene RNA and 18S rRNA was detected on the CFX96 Touch Real-Time PCR Detection System (Bio-Rad, Hercules, CA, USA) using the SuperScript III Platinum One-Step qRT-PCR Kit (Thermo Fisher Scientific) with amplification primers and reporter probes targeting the SOSV N (nucleocapsid) gene [1] and mammalian 18S rRNA gene (Catalog # 4319413E; Applied Biosystems, Grand Island, NY, USA).

### 2.4. Serology

Serum samples were tested at the CDC for the presence of SOSV-specific IgG antibodies by enzyme-linked immunosorbent assay (ELISA) using 96-well plates coated with 100 µL of the dilution of SOSV antigen lysate (diluent: PBS containing 1% thimerosal) that was found to result in optimal reactivity (1:500 dilution) when tested against pooled SOSV bat antisera (*n* = 3) from a previous experimental infection study [13] and pooled SOSV-naïve bat sera (*n* = 19) from an ERB breeding colony. Corresponding plate wells were coated with an equivalent dilution of uninfected control lysate and incubated overnight at 4 °C. Plates were then washed with PBS containing 0.1% Tween-20 (PBS-T). Next, 100 µL of serum diluent (PBS containing 5% skim milk and 0.1% tween-20) was added to each well of the plate. After 10 min, 33 µL of a 21:521 dilution of gamma-irradiated bat serum (dose-2.0 megarads) pre-diluted in masterplate diluent (PBS containing 5% skim milk powder, 0.5% tween-20 and 1% thimerosal) was added to the first well of the plate and four-fold serial dilutions were performed. Final bat serum concentrations were 1:100, 1:400, 1:1600, and 1:6400. Following a 1 h incubation at 37 °C, the plates were washed with PBS-T and 100 µL of a 1:11,000 dilution of goat anti-bat IgG conjugated to horseradish peroxidase (Bethyl Laboratories, Montgomery, TX, USA, Cat#: A140-118P, Lot#: A140-118P-17) in serum diluent was added to the plates. The manufacturer notes that this antibody reacts specifically with bat IgG and with light chains common to other immunoglobulins. After incubation for 1 h at 37 °C, the plates were washed with PBS-T, 100 µL of the Two-Component ABTS Peroxidase System (KPL, Gaithersburg, MD, USA) was added, and the plates were allowed to incubate for 30 min at 37 °C. The plates were then read on a microplate spectrophotometer set at 410 nm. The adjusted optical density (OD) values of each four-fold serial dilution were visually inspected to ensure linearity. To negate non-specific background reactivity, OD values were calculated by subtracting the OD values at each four-fold dilution of wells coated with uninfected control antigen lysate from OD values at corresponding wells coated with SOSV antigen lysate. The adjusted sum OD value was determined by summing the adjusted OD values at each four-fold serial dilution. The mean and standard deviation (SD) of the adjusted sum OD values of 19 ERBs from the breeding colony were used to plot a frequency distribution and calculate a value greater than the mean +3 SD. If a bat had an adjusted sum OD ≥ 0.92, confidence was >99.7% that it was infected with SOSV and had seroconverted.

### 2.5. Virus Isolation 

All virus isolations were performed at the CDC under biosafety level 4 conditions following methods reported previously [13]. Select tissues (pooled liver–spleen, axillary lymph node, and salivary gland) were homogenized in 500 µL DMEM/fungizone/penstrep (100 units/mL penicillin; 100 µg/mL streptomycin; 2.50 µg/mL amphotericin B; Life Technologies) with 2% fetal calf serum and then centrifuged for 10 min at 300× *g*. The entire supernatant was used to inoculate Vero-E6 cells in 25 cm^2^ flasks for 1 h at 37 °C and 5% CO_2_. Maintenance media (DMEM containing 2% fetal bovine serum, 100 units/mL penicillin, and 100 µg/mL streptomycin) was then added to cultures; cells were monitored for 14 days with a media change on day 7. As described previously [13], all cultures were tested by immunofluorescence assay for SOSV antigen at 7- and 14-days post infection.

### 2.6. Sequencing 

SOSV-specific enrichment oligos (Appendix A) were generated using in-house scripts (GitHub-evk3/Nipah_phylogenetics: Collection of scripts used for “Inference of Nipah virus Evolution, 1999–2015”). Briefly, the SOSV (NC_025343) and Tuhoko (NC_025350) reference genomes were parsed to generate 80 bp oligonucleotides that were tiled with no overlaps across both genomes. DNase-treated (Catalog #04716728001, Roche, Basel, Switzerland) RNA extracted from the axillary lymph nodes (ALN) of bat 1021 were prepared for unbiased next generation sequencing (NGS) using the TruSeq RNA Exome Library preparation kit (Illumina, San Diego, CA, USA) and the SOSV-specific enrichment oligos, using a previously established approach [25]. The indexed and pooled libraries were then sequenced using the MiniSeq High-Output Kit (150 cycles; Illumina, San Diego, CA, USA) on the MiniSeq (Illumina). After importing the paired-end sequence reads into Geneious Prime v 2021.0.3 (Biomatters, Auckland, New Zealand) duplicate sequences were removed using Dedupe (Kmer seed length 31) and primer sequences, adaptor sequences and low-quality reads were trimmed using BBDuk. The remaining reads were mapped to the SOSV reference sequence (NC025343) using the Geneious assembler default settings.

### 2.7. Phylogenetic Analysis

Following previously published work [1,2], an outgroup representative of Rubulavirus (Mumps orthorubulavirus, NC_002200) and ingroup taxa characteristic of Rubulavirus-like viruses (Achimota virus 1, NC_025403; Achimota virus 2, NC_025404; Menangle virus, NC_039197; Tioman virus, NC_004074; Tuhoko virus 1, NC_025410; Tuhoko virus 2, NC_025348; Tuhoko virus 3, NC_025350) were used in the analyses, including 13 sequences of the Sosuga virus. Of these, only 1 was obtained from a human host (NC_025343) and the other 12 were obtained from ERBs (*Rousettus aegyptiacus*, Bat-1021 and KP150637–KP150651). It is important to note that four individuals each possessed sequences of both the nucleoprotein and hemagglutinin–neuraminidase genes that were concatenated into one sequence (415 bp). Therefore, the complete dataset contained 21 sequences and a multiple sequence alignment (17,468 bp) was generated using Clustal Omega 1.2.2 [26]. The evaluation of the full distance matrix for genetic similarity and the number of nucleotide differences among sequences of Sosuga virus were estimated using Geneious Prime 2023.2.1 (Geneious, Boston, MA, USA).

A total of 88 maximum likelihood (ML) models were evaluated using jModelTest-2.1.10 [27,28]. The Akaike information criterion with a correction for finite sample sizes [29,30] identified the general time reversible plus proportion of invariable sites plus the gamma distribution model of nucleotide substitution (GTR + I + Γ, −lnL = 148,094.1829) [31] as the most appropriate for the dataset. A maximum likelihood analysis was performed using RAxML (version 8.2.12) [32] with the GTR + I + Γ model of nucleotide substitution and the following parameters for base frequencies: A = 0.3257, C = 0.2149, G = 0.1955, and T = 0.2639. Nodal support was evaluated using the bootstrap method (1000 iterations) [33]. Bootstrap values (BS) ≥ 65 were used to indicate moderate-to-strong nodal support.

A Bayesian inference model (MrBayes v3.2.6) [34] was conducted to generate posterior probability values (PPV). The best-fit model of nucleotide substitution (GTR + I + Γ) and the following parameters were used: two independent runs with four Markov chains (one cold and three heated; MCMCMC), 10 million generations, and sample frequency of every 1000 generations from the last 9 million generated. A visual inspection of the likelihood scores resulted in the first 1 million trees being discarded (10% burn-in) and a consensus tree (50% majority rule) being constructed from the remaining trees. Posterior probability values ≥ 0.95 were used to designate nodal support [35].

## 3. Results

### 3.1. Sosuga Virus qRT-PCR

A total of 377 ERBs were captured at two locations (Kasewe Cave; *n* = 374, and Tailu Village; *n* = 3; Figure 1) and tested for SOSV RNA. Of the ERBs captured and tested, 26.0% (98/377) had detectable viral RNA by virus-specific qRT-PCR, indicating SOSV infection (Table 1; see Appendix A for a list of bat species that tested negative for SOSV RNA). Samples positive for SOSV RNA included oral swabs and visceral tissues, including the liver/spleen, axillary lymph node, salivary gland, and colon (Table 2). Analysis of the ERB demographics with respect to SOSV infections using a Pearson’s chi-squared test of independence indicated a significant age bias in SOSV infections with 31.85% (50/157) of juvenile ERBs (forearm length < 90 mm) [19] being actively infected compared to 21.82% (48/220) of adult ERBs (χ^2^ [1, *n* = 377] = 4.79, *p* = 0.029.). There was no significant bias in SOSV active infection between males and females (χ^2^ [1, *n* = 377] = 0.634, *p* = 0.43) or between adult only males and females (χ^2^ [1, *n* = 172] = 1.601, *p* = 0.206) (Table 1). Pearson’s chi-squared test of independence indicated that there was also no significant difference between age and sex with respect to the total captures tested for SOSV RNA (χ^2^ [1, *n* = 377] = 0.426, *p* = 0.514), eliminating the influence of a trapping bias. 

### 3.2. Serology

A total of 281 ERB blood samples were tested for the presence of SOSV IgG antibodies (Table 1). Of these, 37.72% (106/281) had antibody reactivity against SOSV. Pearson’s chi-squared test of independence indicated a significant age bias with 43.79% of adults (67/153 total adults) having antibody reactivity against SOSV compared to 30.47% (39/128 total juveniles) of juvenile ERBs (χ^2^ [1, *n* = 281] = 5.265, *p* = 0.022). There was also a significant sexual bias in antibody reactivity against SOSV with 45.86% of females (61/133 total females) having antibody reactivity against SOSV compared to 30.41% (45/148 total males), (χ^2^ [1, *n* = 281] = 7.126, *p* < 0.01) of male ERBs. Further analysis with Pearson’s chi-squared test of independence of sex and SOSV antibody layered by age revealed that there were significantly more juvenile female bats (41.545%; 27/65) with antibody reactivity against SOSV than male juvenile bats (19.05%; 12/63; χ^2^ [1, *n* = 128] = 7.6395, *p* < 0.01). Interestingly, a total of 19 juvenile bats that had detectable SOSV RNA also had antibody reactivity against SOSV. There was no significant difference between age and sex with respect to total number of bats tested for antibody reactivity against SOSV (χ^2^ [1, *n* = 281] = 1.123, *p* = 0.289). There was also no significant difference between sexes of adult male and female ERBs with respect to the antibody reactivity against SOSV (χ^2^ [1, *n* = 153] = 1.917, *p* = 0.166), despite the percentage of antibody positive females being higher than males (Table 1).

### 3.3. Virus Isolation

A total of 42 SOSV RNA positive tissue and swab samples from 33 bats underwent virus isolation attempts. Tissues with a qRT-PCR C_T_ value ≤ 35 were selected for isolation. All attempts at isolating infectious SOSV from wild-caught ERBs were unsuccessful.

### 3.4. Sequencing and Phylogenetic Analysis

Sequencing was performed on RNA extracted from an ALN tissue sample that had a relatively low C_T_ value. Approximately 98% (15,269 bp) of the SOSV genome was generated for the Sierra Leone SOSV sequence (Bat-1021, GenBank numbers shown in Appendix A) using the original extracted RNA and 80 bp oligonucleotides that were tiled with no overlaps across both genomes. Maximum likelihood and Bayesian inference analyses generated similar phylogenetic topologies using SOSV bat sequences from Sierra Leone (*n* = 1, 15,269 bp), previously reported sequences from Uganda [*n* = 4 HN and NP gene (415 bp), *n* = 7 NP gene (127 bp) [2], and one previously reported human-derived sequence (15,480 bp) [1]; therefore, only the topology obtained from the Bayesian analysis is shown with the bootstrap and posterior probability values superimposed on each supported node (Figure 2). This analysis produced one major clade of SOSV that contained Bat-1021 from Sierra Leone with the other 11 Ugandan bats and one human sample. In both analyses, the associations between all viruses were strongly supported by both bootstrap and posterior probability values. Genetic similarity calculated between the SOSV sequences indicate a near identical similarity (99.0%) between the SOSV sequence from Sierra Leone and the Uganda human-derived sequence (Table 3). The Sierra Leone sequence was also (96.1–98.1%) similar to the Uganda ERB SOSV sequences, taking into account that the model of evolution was comparing only those sequence regions that were present due to the small sequence fragments of the Ugandan ERB SOSV sequences and ignoring the gaps. To further compare the Sierra Leone sequence to the Ugandan sequences, the number of nucleotide differences between the SOSV sequences were calculated resulting in 158 differences between the near full-length Sierra Leone bat sequence and the human sequence and anywhere from 5–8 differences between the Sierra Leone and smaller Uganda sequence fragments (Table 3).

## 4. Discussion

### 4.1. Sosuga Virus Circulation

The results of the surveillance efforts and subsequent testing of samples presented herein demonstrate that SOSV is circulating in populations of ERBs in Sierra Leone. Maximum likelihood and Bayesian inference phylogenies (Figure 2) constructed using a SOSV RNA sequence obtained from a Sierra Leone ERB sample indicates that there is strong support for SOSV identity based on the position in a clade relative to the SOSV sequence obtained from the human SOSV patient and the SOSV sequences obtained from the Ugandan ERBs. One limitation with this analysis is that the SOSV sequences from the Uganda ERBs are only fragments of the NP (127-nucleotide region) and HN (331-nucleotide region) genes. The analysis used to determine the identity and calculate nucleotide differences compared only the existing sections from the different sequence fragments while ignoring gaps and missing data. As previously reported, Uganda ERB sequences were identical (*n* = 8) or differed by only one nucleotide (*n* = 3) from the human SOSV sequence [2], which is 99.0% similar to the Sierra Leone ERB sequences reported herein (Table 3), further substantiating the presence of SOSV circulation in ERB populations in Sierra Leone. The calculated nucleotide differences between the SOSV sequences (Table 3) also rules out any potential contamination resulting in such a high similarity.

### 4.2. Infection Bias

Analysis of the SOSV PCR data revealed an age bias towards juvenile ERBs with respect to SOSV active infection (31.85% for juveniles, 21.82% for adults; Table 1). A Pearson’s chi-squared test of independence indicates that the bias is not due to a trapping anomaly, but these results are not consistent with previously collected SOSV field data from Uganda [2] where no age bias was detected. This age bias in SOSV infection does mirror that of the bias detected in ERBs with respect to MARV in both Sierra Leone and in Uganda, where significantly more juvenile bats were identified with active MARV infection than adults [18,22]. Experimental infections of ERBs with SOSV demonstrated that the virus is shed orally, but predominantly in urine and feces [13,14]. Considering the ERB population roosting dynamics observed at Python Cave in Uganda where juveniles occupy space closer to, if not directly in, crevasses and holes directly on the floor of the cave, it is possible that the juveniles in the Kasewe Cave complex in Sierra Leone also occupy the lower roosting positions (low walls and floor crevasses) where they are exposed to copious amounts of excreted infectious waste material. It is likely that the same mechanisms leading to MARV infection in older juvenile ERBs [18], primarily waning maternal antibody, are in effect with SOSV infections in the ERBs in Sierra Leone. 

An age bias was also identified in SOSV seroprevalence in favor of adults, which have a higher prevalence of antibody (43.79%) compared to the juveniles (30.47%). Considering the age bias in active infection, the higher seroprevalence of adults makes biological sense and indicates some level of long-term antibody stability after infection with SOSV. Additionally, the influx of susceptible juveniles becoming infected leads to re-exposure of adults, thereby maintaining that cohort’s higher SOSV seroprevalence.

In addition to the serological age bias, there was also a bias towards the female bats with respect to SOSV seroprevalence, with 45.86% having antibodies compared to 30.41% of males, despite there being slightly more males tested (*n* = 148) than females (*n* = 133) (Table 1). There were, however, considerably more juvenile females with antibody reactivity against SOSV (*n* = 27) than males (*n* = 12). Further analysis of sex and SOSV antibody status by age class revealed that the number of female juveniles with antibody reactivity against SOSV did influence the bias towards females in the overall analysis of sex and SOSV antibody status, despite the total numbers of juveniles with antibodies reactive with SOSV being significantly less than the adults. All the juveniles appeared to be of the same approximate age based on morphometric measurements. The average forearm length of the juveniles with antibodies reactive against SOSV was 84.5 mm, which puts them loosely into a previously published 4-month-old age category [36]. The caveat with this age category is that it was developed for ERBs in the Mediterranean and Egyptian desert regions where there could be geographically influenced size differences from those juvenile ERBs in West Africa. Nevertheless, this age category corresponds to a period where ERB juveniles in Uganda typically become weened and lose maternal antibodies [19,37] leaving them vulnerable to infection. This waning of maternal antibody and subsequent infection with MARV was observed in Ugandan ERBs [18]. It is possible that the antibody reactivity against SOSV detected in the juvenile ERBs could be residual maternal antibody, but of the 39 juveniles that had antibody reactive against SOSV, 19 of them also had detectable SOSV RNA, indicating that they were actively infected with SOSV after maternal antibodies had waned. Maternal antibodies for other paramyxoviruses like HeV and canine distemper virus are estimated to wane at about 6 months of age in pteropodid bats [38,39] and MARV maternal antibody was experimentally shown to wane at 5 months in ERBs [40]. It is possible that the juveniles with antibody reactivity against SOSV are older than the estimated 4 months and are actually closer to 5–6 months and their maternal protective immunity had waned leaving them susceptible to infection with SOSV. By 6 months of age the juveniles are independent and flying on their own [15], further exposing them to potentially infectious sources.

Analysis of the antibody data indicates a bias towards juvenile females having more antibody reactivity against SOSV than male juveniles. For reasons unclear, this appears to occur despite there being an equal number of juvenile males and females with detectable SOSV RNA (Table 1). It is worth noting that sexual dimorphism with respect to viral infections has been reported to be more male-centric across a variety of taxa due to sex steroid hormones [41]. However, there are exceptions to this with one being measles, a paramyxovirus like SOSV [42]. This could be the reason more females have become infected at an early age and have seroconverted before the males. Naive juvenile females roosting in crevices and holes on the cave floor could be one route of exposure to SOSV, given the amount of observed SOSV shedding by experimentally infected ERBs [13,14]. Another speculative explanation for increased SOSV infections in female juvenile ERBs, other than their roost position in the cave, could be that the juvenile females are being approached by an SOSV-infected adult males looking for a mate, despite the age and sexual immaturity of the females. Both males and females reach sexual maturity at one year [19]. Adult males often bite females to get them to submit to copulation [43] and SOSV has been shown to be shed in oral secretions as well as excreta [13,14]. Given the possibility that females could be more susceptible than males to a paramyxovirus infection, these advances by the adult males could potentially result in an increased active infection rate and subsequent antibody production among juvenile females and produce a true bias in favor of juvenile females over males with respect to antibody reactive against SOSV.

### 4.3. Public Health

The discovery of SOSV in ERB populations in Sierra Leone represent an increased risk to public health in the event of contact with these bats or their excreta. Fortunately, like the discovery of MARV in these bats in Sierra Leone [22], SOSV was identified before any outbreak of disease that will enable preemptive safety messaging to the public about avoiding contact with these bats. Moreover, caution must be taken with respect to agricultural and wild harvesting of fruits in areas surrounding Kasewe Cave or other known ERB roosting sites. GPS tracking of ERBs in Uganda has shown that these bats will fly considerable distances to forage on human cultivated fruits where they may deposit zoonotic pathogens on uneaten and dropped fruits through their saliva, urine, and feces [44]. Similar to the public messaging with ERBs and MARV in Sierra Leone, the best practice for ensuring public health is avoidance of ERBs and their roosting sites.

### 4.4. Reservoir Status 

The field data presented herein, coupled with results of other field studies [2] and experimental infections of ERBs with SOSV where subclinical systemic infections occurred, and viral shedding identified [13,14] have indicated that ERBs are a potential reservoir for SOSV. Finding SOSV in ERB populations in Sierra Leone so far removed from the SOSV RNA positive bats in Uganda support the notion that these common African fruit bats are at the very least competent SOSV amplification hosts. More data, specifically multiple isolates obtained from wild caught ERBs over extended time periods and multiple locations, would solidify the SOSV reservoir status of these bats, which are already a known reservoir for MARV and potential reservoirs for multiple other rubulaviruses [3,17,18,21]. These findings also represent a substantial range extension from East Africa to West Africa for SOSV, suggesting that this paramyxovirus may occur in ERB populations throughout its sub-Saharan African distribution.

## Figures and Tables

**Figure 1 viruses-16-00648-f001:**
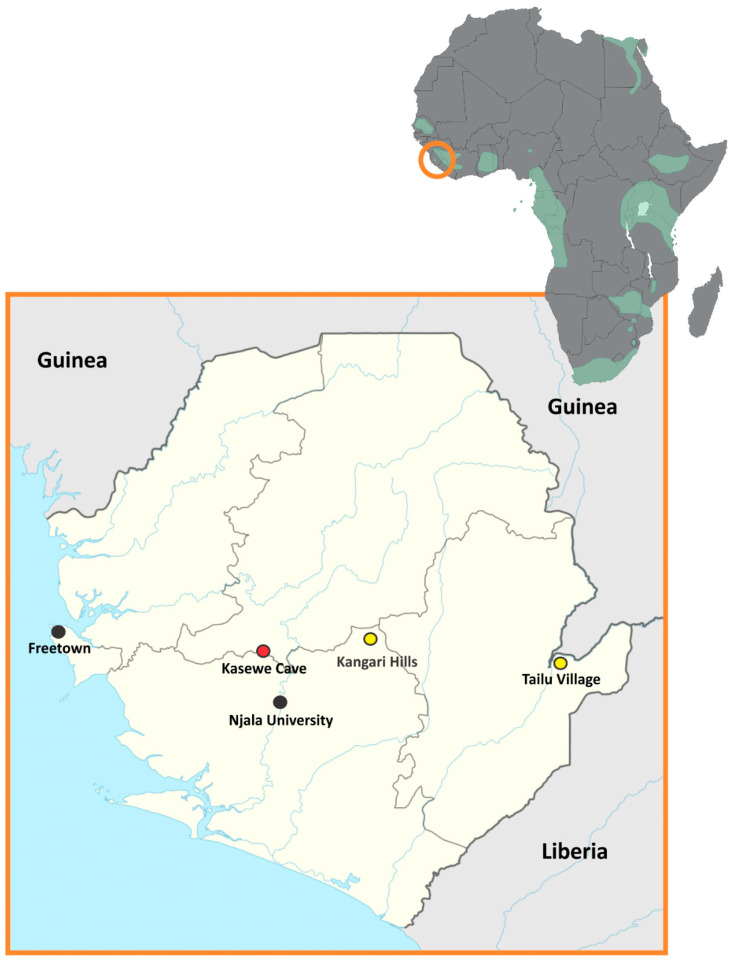
Distribution of Egyptian rousette bats (ERBs; *Rousettus aegyptiacus*) in Africa (upper right corner) with Sierra Leone highlighted by orange circle. Map of Sierra Leone (lower left corner) showing zoonotic surveillance trapping sites (red and yellow dots), Njala University and Freetown (black dots).

**Figure 2 viruses-16-00648-f002:**
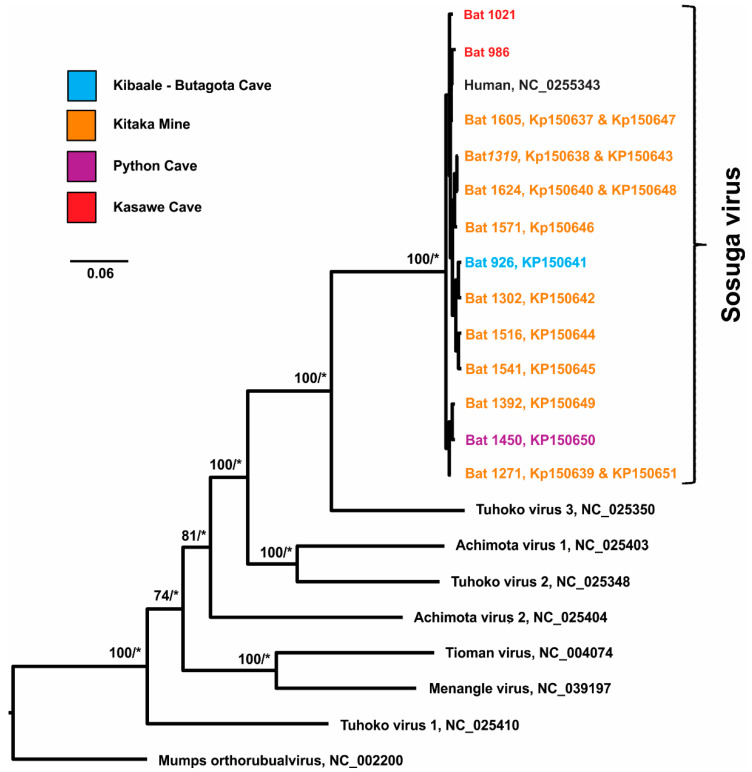
Phylogeny of Sosuga virus (SOSV) using 13 individual bats (*Rousettus aegyptiacus*) and one human host. Indicated on the nodes are maximum likelihood bootstrap values representing ≥ 65 nodal support (number left of the slash) and Bayesian posterior probability values indicated by the asterisk (*) representing ≥0.95 nodal support (right of the slash). Sequence lengths are as follows: Human,: 15,480 bp, Bat-1021 (Sierra Leone): 15,269 bp, Bat-1271 (HN and NP genes): 415 bp, Bat-1319 (HN and NP genes): 415 bp, Bat-1624 (HN and NP genes): 415 bp, Bat-1605 (HN and NP genes): 415 bp, Bat-1392 (NP gene): 127 bp, Bat-1302 (NP gene): 127 bp, Bat-1516 (NP gene): 127 bp, Bat-1541 (NP gene): 127 bp, Bat-1571 (NP gene): 127 bp, Bat-926 (NP gene): 127 bp, Bat-1450 (just NP gene): 127 bp (see Appendix A for a list of GenBank numbers).

**Table 1 viruses-16-00648-t001:** Summary of *Rousettus aegyptiacus* captures displayed by sex and age class, Sosuga virus (SOSV) RNA status, and anti-SOSV IgG status.

		*n*	SOSV RNA+	(%)	*n*	Anti-SOSV IgG+	(%)
Adult	Female	106	27	25.47	68	34 **	50.0
	Male	114	21	18.42	85	33	38.82
	Total	220	48	21.82	153	67	43.79
Juvenile	Female	81	25	30.86	65	27 **	41.54
	Male	76	25	32.90	63	12	19.05
	Total	157	50	31.85	128	39	30.47
Total		377	98	26.00	281	106 *	37.72

* SOSV serology *n* = 281 bats tested. ** Shows sexual bias towards females with 45.86% (61/133) identified as having antibodies reactive against SOSV compared to 30.41% (45/148) of males.

**Table 2 viruses-16-00648-t002:** Sosuga virus (SOSV) RNA cycle threshold (C_T_) minimum and maximum values detected by qRT-PCR in *Rousettus aegyptiacus* tissues in Sierra Leone.

SOSV	Oral Swab	Liver/Spleen	Axillary Lymph Node	Salivary Gland	Colon/Rectum
Number positive	2	39	85	28	19
C_T_ Min	27.31	32.13	27.32	31.62	29.11
C_T_ Max	37.13	39.88	39.68	39.57	39.02

**Table 3 viruses-16-00648-t003:** A pairwise comparison of Sosuga virus (SOSV) sequence identity (top) and nucleotide differences (bottom), calculated using Geneious Prime, for the Sierra Leone (B1021), Uganda, and Human SOSV sequences. The analysis used to determine the identity and calculate nucleotide differences compared only existing sections from the different sequence fragments (127 bp) while ignoring gaps and missing data.

	Human	B1021	B1271	B1319	B1605	B1624	B926	B1302	B1392	B1450	B1516	B1541	B1571
Human		98.97	99.04	99.28	99.52	99.28	100.00	100.00	99.21	99.21	100.00	100.00	100.00
B1021	158		98.07	98.07	98.31	98.07	96.06	96.06	96.06	96.06	96.06	96.06	96.06
B1271 *	4	8		98.8	99.04	98.8	99.21	99.21	100.00	100.00	99.21	99.21	99.21
B1319 *	3	8	5		99.28	100.00	100.00	100.00	99.21	99.21	100.00	100.00	100.00
B1605 *	2	7	4	3		99.28	100.00	100.00	99.21	99.21	100.00	100.00	100.00
B1624 *	3	8	5	0	3		100.00	100.00	99.21	99.21	100.00	100.00	100.00
B926 **	0	5	1	0	0	0		100.00	99.21	99.21	100.00	100.00	100.00
B1302 **	0	5	1	0	0	0	0		99.21	99.21	100.00	100.00	100.00
B1392 **	1	5	0	1	1	1	1	1		100.00	99.21	99.21	99.21
B1450 **	1	5	0	1	1	1	1	1	0		99.21	99.21	99.21
B1516 **	0	5	1	0	0	0	0	0	1	1		100.00	100.00
B1541 **	0	5	1	0	0	0	0	0	1	1	0		100.00
B1571 **	0	5	1	0	0	0	0	0	1	1	0	0	

* Bat, Uganda (HN and NP gene): 415 bp. ** Bat, Uganda (NP gene): 127 bp. Human, Uganda: 15,480 bp. B1021, Sierra Leone: 15,269 bp.

## Data Availability

The authors declare that all the data supporting the findings of this study are available within the article and/or from the authors upon request.

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
