# Peer review of "Sosuga Virus Detected in Egyptian Rousette Bats (Rousettus aegyptiacus) in Sierra Leone"

_viruses, 2024, doi:10.3390/v16040648_

Round 1

Reviewer 1 Report

Comments and Suggestions for Authors

The authors describe the results of surveillance for Sosuga virus (SOSV) in Egyptian rousette bats in Sierra Leone. The detection of SOSV outside of Uganda significantly increases the known range of this zoonotic virus. While the manuscript is generally straightforward, I have major concerns regarding data availability and transparency. Data sharing is now a standard expectation in the scientific community, and I am extremely disappointed to see the lack of data availability and transparency in this manuscript. The statement “data upon request” is notoriously unreliable, harmful to the broader scientific community and to the longevity of scientific data and is not acceptable for modern published works. This is particularly important for data generated on public funds as this work has been.

Most useful would be depositing surveillance data in a searchable repository such as the newly established PHAROS (https://pharos.viralemergence.org/). In this case the sampling date and coordinates, species, age, and sex with corresponding PCR and serological assay data are the minimum requirements for data transparency. The authors can include this as an excel table in the supplemental, deposit raw data to an appropriate searchable repository (e.g. PHAROS), or deposit in a data server (e.g. Dryad). In addition, raw data from ELISA and sequencing data are needed.

Sequencing:
The reference here (25) is for CCHFV, not SOSV. Please include enrichment oligos designed for SOSV – this is basic information for reproducibility and transparency. Withholding this information not only reduces the likelihood that this work will be cited, but will force others to needlessly duplicate the effort.  

Minor concerns listed below:

Abstract:
Line 30-31: Please rephrase, describes one-to-one relationship between SOSV and ERBs as reality/truth rather than observations. Observations are subject to sampling bias and incomplete sampling.  

Introduction:
Line 73-74: The authors are suggesting that the sampling of other taxa in the region implicates ERBs as the sole host of SOSV. This is an interesting thought which would worth further statistical analyses. Given the prevalence of SOSV RNA detection in ERBs, what is the probability of missing SOSV RNA in other species given the depth of sampling and screening for each of those 50 taxa?

Very minor formatting issue: Please do not alternate between double space and single space after each period. Just use single space.

Author Response

Reviewer 1.

The authors describe the results of surveillance for Sosuga virus (SOSV) in Egyptian rousette bats in Sierra Leone. The detection of SOSV outside of Uganda significantly increases the known range of this zoonotic virus. While the manuscript is generally straightforward, I have major concerns regarding data availability and transparency. Data sharing is now a standard expectation in the scientific community, and I am extremely disappointed to see the lack of data availability and transparency in this manuscript. The statement “data upon request” is notoriously unreliable, harmful to the broader scientific community and to the longevity of scientific data and is not acceptable for modern published works. This is particularly important for data generated on public funds as this work has been.

Most useful would be depositing surveillance data in a searchable repository such as the newly established PHAROS (https://pharos.viralemergence.org/). In this case the sampling date and coordinates, species, age, and sex with corresponding PCR and serological assay data are the minimum requirements for data transparency. The authors can include this as an excel table in the supplemental, deposit raw data to an appropriate searchable repository (e.g. PHAROS), or deposit in a data server (e.g. Dryad). In addition, raw data from ELISA and sequencing data are needed.

We respectfully disagree with the reviewer’s view that we would not provide the data upon request. We are not sure who the reviewer is referring to, but in the past we have always honored requests for data.  That is why we say "upon request" so we can determine who the requesters are and what they will be doing with that data. We have had our data published before without our consent and are extremely leery of just putting everything into an open databank as requested.

Sequencing:
The reference here (25) is for CCHFV, not SOSV. Please include enrichment oligos designed for SOSV – this is basic information for reproducibility and transparency. Withholding this information not only reduces the likelihood that this work will be cited, but will force others to needlessly duplicate the effort.

We agree with this and acknowledge our oversight. The 381 enrichment oligos designed for SOSV will be provided in a supplemental table. Thank you for pointing this out to us.

Minor concerns listed below:

Abstract:
Line 30-31: Please rephrase, describes one-to-one relationship between SOSV and ERBs as reality/truth rather than observations. Observations are subject to sampling bias and incomplete sampling.

We respectfully disagree with the reviewer’s assessment of this statement of our repeated findings in Uganda that ERBs have been the only species of bat to have tested positive for SOSV. We are not saying that it is the reservoir or that SOSV does not spill over into other species of bats. We are simply sating that based on our data, no other species have been found to be infected with SOSV.  To be more clear, we have changed the sentence to read “Since that time, multiple other species have been sampled and ERBs in Uganda have continued to be the only species of bat positive for SOSV infection.” 

Introduction:
Line 73-74: The authors are suggesting that the sampling of other taxa in the region implicates ERBs as the sole host of SOSV. This is an interesting thought which would worth further statistical analyses. Given the prevalence of SOSV RNA detection in ERBs, what is the probability of missing SOSV RNA in other species given the depth of sampling and screening for each of those 50 taxa?

We do not see this as an implication of ERBs being the sole host to SOSV. The sentence reads “Of all the species tested from this region, only the ERB samples had detectable SOSV RNA. This is the first identified occurrence of SOSV in bats outside of East Africa (Uganda) and the first reported occurrence of this human pathogenic paramyxovirus in West Africa (Sierra Leone).” and suggests only that this is the first time we have found SOSV outside of Uganda. In fact, we state on lines 68-72 “Moreover, a recent studies report that captive bred ERBs could be experimentally infected with SOSV with no overt signs of morbidity, mild pathology, and shed the virus in urine, feces, and saliva, suggesting they are at least a competent host and a potential reservoir for this human pathogenic paramyxovirus [13,14].”. We agree that it may be possible to find SOSV infection in another species, as we have seen many times with spillover of Marburg virus to a non-reservoir species. In section 4.3. Reservoir status, (Lines 440-442) we state “The field data presented herein, coupled with results of other field studies [2] and experimental infections of ERBs with SOSV where subclinical systemic infections occurred, and viral shedding identified [13,14] have indicated that ERBs are a potential reservoir for SOSV.”.

Very minor formatting issue: Please do not alternate between double space and single space after each period. Just use single space.

Done, thank you.

Reviewer 2 Report

Comments and Suggestions for Authors

General comment:

Well written report on the detection of Sosuga virus (SOSV) in Egyptian Rousette Bats (ERB’s) in Sierra Leone. The authors found SOSV genome in ERB tissue samples and swabs, although no virus was able to be isolated, and SOSV antibodies in ERB serum samples. Phylogenetic analysis showed near-identical similarity to the human isolate from Uganda and very high similarity to the other Ugandan ERB sequences available. Previously SOSV has only been detected in ERB’s in East Africa, so this report extends the range in which this virus can be found to West Africa.

Minor corrections:

Line 59 – Change “Moreover, a recent studies…” to “Moreover, recent studies…”

Line 105 – “Viral transport media” - Please provide details of media used

Line 168 – Change “300 g” to “300 xg

Table 2 – Consider placing / between Liver Spleen and Colon Rectum e.g. Liver/Spleen and Colon/Rectum

Line 397 – Change from “juvenile males and female” to “juvenile males and females”

Line 403 – Change “roosting in cervices” to “roosting in crevices”

Author Response

Reviewer 2.

Comments and Suggestions for Authors

General comment:

Well written report on the detection of Sosuga virus (SOSV) in Egyptian Rousette Bats (ERB’s) in Sierra Leone. The authors found SOSV genome in ERB tissue samples and swabs, although no virus was able to be isolated, and SOSV antibodies in ERB serum samples. Phylogenetic analysis showed near-identical similarity to the human isolate from Uganda and very high similarity to the other Ugandan ERB sequences available. Previously SOSV has only been detected in ERB’s in East Africa, so this report extends the range in which this virus can be found to West Africa.

Minor corrections:

 Line 59 – Change “Moreover, a recent studies…” to “Moreover, recent studies…” – Done, thank you.

Line 105 – “Viral transport media” - Please provide details of media used - Done, thank you.

Line 168 – Change “300 g” to “300 xg” – Done, thank you.

Table 2 – Consider placing / between Liver Spleen and Colon Rectum e.g. Liver/Spleen and Colon/Rectum - Done, thank you.

Line 397 – Change from “juvenile males and female” to “juvenile males and females” - Done, thank you.

Line 403 – Change “roosting in cervices” to “roosting in crevices” - Done, thank you.

Reviewer 3 Report

Comments and Suggestions for Authors

Minor editing comments:

Line 47 – Reference repeated twice

Line 59 – Remove “a” as referring to multiple studies.

Line 135 – Change “plates” to “plate”

Line 156 – remove “virus” after SOSV

Line 181 – the word bat is duplicated

Line 200 – Rousettus aegyptiacus should be in italics

Line 240 – Remove additional full stop after p value

Line 251 and 252 – spacing

Line 266 - Change %19.05 to 19.05% and remove the additional full stop after the p-value

Comments:

Line 165 – Only referring to "select tissues homogenized" – please specify the tissue types from which isolation was attempted

Section 2.1 - It would be beneficial to add the timing of sampling. What time of year were samples collected?

Line 231 - Was all sample types that were collected tested for all bat species? This was not clear.

Author Response

Reviewer 3.

Comments and Suggestions for Authors

Minor editing comments:

Line 47 – Reference repeated twice – Extra reference deleted, thank you.

Line 59 – Remove “a” as referring to multiple studies. - Done, thank you.

Line 135 – Change “plates” to “plate” - Done, thank you.

Line 156 – remove “virus” after SOSV - Done, thank you.

Line 181 – the word bat is duplicated – Extra bat deleted, thank you.

Line 200 – Rousettus aegyptiacus should be in italics - Formatting corrected, thank you.

Line 240 – Remove additional full stop after p value - Done, thank you.

Line 251 and 252 – spacing – Corrected, thank you.

Line 266 - Change %19.05 to 19.05% and remove the additional full stop after the p-value - Done, thank you.

Comments:

Line 165 – Only referring to "select tissues homogenized" – please specify the tissue types from which isolation was attempted – The sentence was changed to read “Select tissues (pooled liver-spleen, axillary lymph node, and salivary gland) were homogenized in 500 µL…”

Section 2.1 - It would be beneficial to add the timing of sampling. What time of year were samples collected? – We agree. The sentence “The majority of bat captures occurred during the fall (September – November, 2017-2020) with one capture event occurring in February 2021.” was added at line 108 in section 2.1.

Line 231 - Was all sample types that were collected tested for all bat species? This was not clear. – Yes, tissue sampling was the same across species captured. Yes, the sentence at lines118- 119 was changed to read “Visceral tissues (liver, spleen, axillary lymph node, salivary gland, and colon) were collected from all captured bat species during necropsies and either flash frozen …:” and the sentence “No bat species captured and sampled in this study were classified as either threatened or endangered.” was added to the end of the paragraph at line 121.

Round 2

Reviewer 1 Report

Comments and Suggestions for Authors

The authors refuse to share the relevant Sosuga virus surveillance data, despite that the data have been generated on public funds and do not contain any sensitive patient or clinical data. This is in contrast with scientific norms, prohibits reuse or data aggregation following publication, and is inconsistent with the FAIR principles under which this and other reputable journals operate. My recommendation is to reject this manuscript unless the authors share minimal raw data from their surveillance efforts. 

The authors may opt out of a more collaborative, open, and searchable database, but the authors are not in compliance with FAIR principles unless they deposit appropriate raw data in a findable repository such as Dryad or Figshare. 

According to MDPI: 
MDPI is committed to supporting open scientific exchange and enabling our authors to achieve best practices in sharing and archiving research data. We encourage all authors of articles published in MDPI journals to share their research data including, but not limited to protocols, analytic methods, raw data, processed data, code, software, algorithms, and study material. The data should be FAIR – findable, accessible, interoperable, and reusable – so that other researchers can locate and use the data.

Author Response

Round 2:

Comments and Suggestions for Authors
The authors refuse to share the relevant Sosuga virus surveillance data, despite that the data have been generated on public funds and do not contain any sensitive patient or clinical data. This is in contrast with scientific norms, prohibits reuse or data aggregation following publication, and is inconsistent with the FAIR principles under which this and other reputable journals operate. My recommendation is to reject this manuscript unless the authors share minimal raw data from their surveillance efforts.

The authors may opt out of a more collaborative, open, and searchable database, but the authors are not in compliance with FAIR principles unless they deposit appropriate raw data in a findable repository such as Dryad or Figshare.

We have never refused sharing our data with anyone. We simply ask that they contact us and ask for the data.

According to MDPI:
MDPI is committed to supporting open scientific exchange and enabling our authors to achieve best practices in sharing and archiving research data. We encourage all authors of articles published in MDPI journals to share their research data including, but not limited to protocols, analytic methods, raw data, processed data, code, software, algorithms, and study material. The data should be FAIR – findable, accessible, interoperable, and reusable – so that other researchers can locate and use the data.

We would like to point out that the other 3 articles in the Special Issue Bat- and Rodent-Borne Zoonotic Viruses have data availability statements like ours.

Hemnani et al. 2024

Presence of Alphacoronavirus in Tree- and Crevice-Dwelling Bats from Portugal

Data Availability Statement

Data are contained within the article.

Makenov et al. 2023

 Detection of Filoviruses in Bats in Vietnam

Data Availability Statement

The data presented in this study are available on request from the corresponding author

Hemnani, et al. 2024

First Report of Alphacoronavirus Circulating in Cavernicolous Bats from Portugal

Data Availability Statement

The data presented in this study are available on request from the corresponding author.
